# Understanding satisfaction and dissatisfaction of patients with telemedicine during the COVID-19 pandemic: An exploratory qualitative study in primary care

**Karolina Pogorzelska** [1] *, **Ludmila Marcinowicz** [2], **Slawomir Chlabicz** [1]

**1** Department of Family Medicine, The Medical University of Bialystok, Bialystok, Poland, **2** Department of Obstetrics, Gynecology and Maternity Care, The Medical University of Bialystok, Bialystok, Poland

* karolina.pogorzelska@sd.umb.edu.pl

## Abstract

### Background

Due to the COVID-19 pandemic, healthcare organizations had to face challenging circumstances and modify the usual modality of service provision, introducing telehealth services in their routine patient care to lessen the risk of direct human-to-human exposure. Patients expressed concerns about personal visits to healthcare units and the possibility of accessing telemedicine turned out to be an effective tool for the continuity of care. Due to the limited experience with telemedicine before the COVID-19 pandemic in Poland, we sought to fill this gap by studying the experiences of Polish patients. Our study aimed to understand how patients define satisfaction and dissatisfaction with telemedicine during the COVID-19 pandemic in primary care.

### Material and methods

Twenty semi-structured interviews with primary care patients in the Podlaskie Voivodeship, Poland were conducted to understand satisfaction with telemedicine.

Interview transcripts were analyzed using qualitative content analysis. The qualitative content analysis process involved familiarizing ourselves with the data, extracting text regarding satisfaction and dissatisfaction with the teleconsultation, condensing it into meaningful units assigning codes to them, and organizing codes into subcategories and categories. The entire analysis process was done through reflection and discussion until a consensus was reached between the researchers.

### Results

From the participants' perspective, satisfaction with telemedicine was associated with receiving enough space to express their concerns. It was reported that they trusted their primary care physicians and felt comfortable during telemedicine consultations. Participants noted that connecting with a known, trusted doctor was more important than having a face-to-face visit with an unfamiliar physician. In our study, the participants equated satisfaction

**Data Availability Statement:** Data cannot be shared publicity as participants did not give consent for their transcripts to be shared in the

public domain. Data are available for selected researchers from the Medical University of Bialystok, who meet the criteria for access to confidential data. Requests for access to the underlying data may be directed to the Bioethics Committee of the Medical University of Bialystok via email (komisjabioetyczna@umb.edu.pl).

**Funding:** This research had financial support from the Medical University of Bialystok, Poland (B. SUB.23.113). The funders had no role in study design, data collection and analysis, decision to publish, or preparation of the manuscript.

**Competing interests:** The authors have declared that no competing interests exist.

with treatment effectiveness. It was emphasized that in the event of unknown or unstable conditions, patients would prefer to be seen in person and receive a physical examination.

## Conclusion

In our research telemedicine met with a positive reception and was recognized by the majority of patients who made use of it as a valuable channel of contact with a primary care physician. In order to increase the level of patient satisfaction, the focus should be on improving aspects such as physician engagement and showing empathy during telemedicine, as well as providing complete, exhaustive information on the treatment process. Respecting patient needs and preferences during performing telemedicine visits is the goal of patient-centered care.

## Background

The World Health Organization (WHO) on March 11, 2020, declared coronavirus infectious disease 2019 caused by SARS CoV-2 virus infection, as a global pandemic that would affect all countries of the world [1]. The lack of specific treatments forced governments to focus on preventive strategies, such as lockdowns, social distancing, and quarantine which have been adopted to reduce viral transmission [2]. Due to the COVID-19 pandemic, healthcare organizations had to face challenging circumstances and modify the usual modality of service provision and introduce telehealth services in their routine patient care to lessen the risk of direct human-to-human exposure [3]. Patients expressed concerns about personal visits to healthcare units and the possibility of obtaining telemedicine turned out to be an effective tool for the continuity of care [4, 5]. Telemedicine had been gradually implemented for approximately a decade, and the COVID-19 pandemic resulted in a large-scale expansion of telemedicine use throughout healthcare facilities and primary care practices all over the world [6, 7].

The most common definition of telemedicine is provided by the World Health Organization WHO: "The delivery of healthcare services, where distance is a critical factor, by all healthcare professionals using information and communication technologies for the exchange of valid information for diagnosis, treatment, and prevention of disease and injuries, research and evaluation, and for the continuing education of healthcare providers, all in the interests of advancing the health of individuals and their communities [8]. In Poland, telemedicine is a relatively new healthcare delivery tool.

The report published at the turn of 2016 and 2017 indicated that only 7% of the Poland population (out of 38.5 million) uses medical services via the Internet. The average in the European Union countries is 13%. The report indicated that 98% of the population has no concerns about e-privacy, and 48% of the population seeks information about their health via the Internet [9]. According to the data for 2018, collected by Statistics Poland, 83% of all Polish households had broadband internet access at home. This has grown to 92,6% in 2022 due to the COVID-19 pandemic [10].

In order to develop the digitalization of the health sector in Poland, many projects supervised by the Minister of Health were adopted and successively implemented.

The first project implemented in 2018 'Electronic Platform for Collection, Analysis and Sharing of Digital Resources on Medical Occurrences' provide the possibility of issuing and using e-prescriptions and e-referrals, and introduced the Patient Internet Account (IKP). By setting up an account with IKP and then logging in, patients have access to their medical records. The next project, Electronic Medical Records (EDM) involved the digitalization of

medical records in accordance with the CDA HL 7 standard. EDM include information on diagnosis, results, recommendations or refusals of admission to hospital, e-prescriptions and e-referrals, as well as an information card covering the entire process of inpatient treatment [11]. 'Improving the quality of management in healthcare by the popularization of knowledge about ICT" was an educational project completed in 2015, in which employees acquire adequate professional knowledge of the use of modern technologies in the workplace [11]. The aim of the 'Reducing social inequalities in health through the use of telemedicine and e-health solutions' telemedicine project was to reduce the costs of medical procedures and facilitate access to medical services for patients in several areas of medicine: cardiology, geriatrics, obstetrics, palliative care, chronic diseases, diabetes, and psychiatry [11].

Legally providing healthcare services remotely started to be possible at the end of 2015 pursuant to Article 3, paragraph 1 of the Act on Medical Activity [12]. Although telemedicine visits were introduced into the system within primary care on 5 November 2019 by decree of the Minister of Health, the actual number of remote consultations between physicians and patients was rather low [13]. The announcement of the state of the COVID-19 pandemic and the need to provide health services safely for the patient and medical staff resulted in telemedicine being implemented in March 2020 on a large scale in primary health care [14]. In March 2021, the Ministry of Health issued a legal act regulating telemedicine visits. Primary care physicians could not refuse a face-to-face visit to a patient who refused remote consultations and had no suspicion of a COVID-19 infection. According to the report published in July 2020, by the Polish National Health Fund, more than 80% of all visits to primary care physicians from the beginning of the COVID-19 pandemic were carried out remotely, mainly by telephone [14]. The vast majority of patients experienced telemedicine consultations for the first time.

In 2021, the Polish National Health Fund introduced the First Contact Teleplatform with one toll-free number for the citizens of the entire country. Medical advice is provided not only in Polish but also in English, Ukrainian, and Russian, as well as in sign language via video chat. The services of the First Contact Teleplatform can be used outside the working hours of primary healthcare facilities. [15] The view that telemedicine is the future of primary health care was well established before the pandemic, however, it was greatly enhanced by the considerable benefits revealed during the COVID-19 era [16, 17]. The importance of telemedicine in primary care extends beyond the COVID-19 pandemic. It provides convenient access to healthcare, especially for those with limited in-person options. This accessibility bridges healthcare gaps, ensuring a wider population can receive timely medical guidance. 61,9% of all Polish patients received a teleconsultation in 2020 and 2021, which was considerably higher than the EU average of 39%. This is the 3rd highest number in Europe (after Spain and Slovenia) [18]. In 2022, 23.5 million teleconsultations were performed in primary medical care and 10.6 million in specialist care [19]. What is noteworthy, there is a fundamental shortage of doctors in Poland, which has 3,44 practicing doctors per 1000 inhabitants and remains one of the lowest among EU countries [20]. This has an impact on the accessibility of care, mainly in more rural regions. In 2020, 1.9% of the Polish population reported unmet needs for medical examinations due to either costs, distance, or waiting times (the EU average was 1.8%) [21]. Telemedicine facilitates continuous care, monitoring chronic conditions, managing medications, and promoting preventive measures, all while offering convenience. Some 39% of Polish adults reported having at least one chronic disease in 2019 –a slightly higher proportion than across the EU as a whole (36%) [21]. Ongoing technological advancements are improving the quality of telemedicine services, making teleconsultations more interactive and informative.

Nevertheless, the effective implementation of telemedicine into primary health care, particular areas such as patients' experiences needs to be investigated further. Patient experiences are the main factor determining how healthcare meets the patients' needs, and directly leads to

improvements in healthcare services [22, 23]. In the literature, patients' perceptions of tele-medicine are often used as a measurement of patient satisfaction [24]. The importance of patient satisfaction is further demonstrated as fundamental in patient-centered care, which directly improves patients' outcomes [25, 26]. Patient satisfaction is a complex and difficult concept that requires further elucidation [27].

Due to the limited experience with telemedicine before the COVID-19 pandemic in Poland, we sought to fill this gap by studying the experiences of Polish patients. The aim of our study was to understand how patients define satisfaction and dissatisfaction with telemedicine during the COVID-19 pandemic in primary care.

## Material and methods

### Design

An exploratory qualitative study using semi-structured telephone interviews was conducted among patients receiving primary care in the Podlaskie Voivodeship. The interviews were conducted between March 2021 and February 2022. Interviews varied in length between 12 min and 25 min, with an average of 14 min. Field notes were made during each interview. The interviews were performed until the research team agreed that data saturation had been achieved: no new threads in the patients' statements could have been identified, new codes have been no longer identified during initial coding, and agreement that sufficient data had been collected to adequately address the research aims [28].

The reporting of this qualitative study was guided by the Consolidated Criteria for Reporting Qualitative Research (COREQ) guidelines [29] (S1 File).

### Participants

Patients 18 years old or older who had a telemedicine consultation with their existing primary care physician were eligible for inclusion. The participants were recruited from eight primary healthcare clinics at which they were registered as patients. Twenty-four patients were invited to participate in the study over the phone. Prior to the interview, participants received an introductory letter, which provided further information about the aim of the study and the consultation process. Four male patients refused to take part in the interview without giving any reason. Purposeful sampling was used to recruit patients from a range of demographics. None of the interviewees had a prior relationship with the members of the research team.

### Measurements

Individual, telephone semi-structured interviews were carried out according to an interview guide. The interview topic guide included the questions listed in Table 1 below.

**Table 1. Questions included in the interview topic guide.**

| Guide questions |
| :---: |
| What do you think about the teleconsultation you have had? |
| How do you assess the results of the teleconsultation? |
| What went well in your teleconsultation? |
| What could the doctor have done better during the teleconsultation? |
| Tell me if you were able to talk about your problem with the doctor. Please elaborate. |
| Tell me if you were listened to during your teleconsultation. Please elaborate. |
| Tell me why are you satisfied or dissatisfied with your teleconsultation. |

The telephone interviews were performed at a time convenient for each participant. All interviews were audio recorded and then transcribed verbatim. All the interviews were conducted in Polish by the same interviewer (KP). There were no repeat interviews.

## Ethics

Participation in the study was voluntary. Each patient was asked for consent at the beginning of the interview and oral informed consent was obtained from the participants. Transcripts were anonymized to protect the identity of the participants. Interview recordings were stored on an external password-secure hard drive. The study was approved by the Bioethics Committee of the Medical University of Bialystok, Poland (No. APK.002.146.2021).

## Data analysis

Interview transcripts were analyzed using qualitative content analysis described by Graneheim and Lundman [30]. The process of collecting and analyzing data was iterative in nature. Three researchers with different backgrounds were involved in data analysis (KP, a family medicine physician in training and PhD student; LM, a professor and nurse; SC, a professor and family medicine physician). Two researchers (LM and SC) were experts in qualitative research, with numerous publications in peer review journals.

The principal investigator (KP) prepared the qualitative data, including the transcripts, following each interview. All researchers read and re-read the transcripts to gain a sense of the content and overview of the material. Two research team members extracted the text regarding satisfaction and dissatisfaction with the teleconsultation, and then condensed them into meaningful units and assigned codes to them. A co-investigator (LM) independently read the interview transcripts and analyzed them. Both researchers consistently compared codes and units of meaning to identify similarities and differences, with the aim of being as internally homogeneous and externally heterogeneous as possible. They revisited the raw data whenever necessary. The codes were then sorted into subcategories and categories. The entire process of analysis was done through reflection and discussion until a consensus was reached between the two researchers. In the case of disagreement, the third researcher was consulted. The final categories were then discussed among the research team members. Coding and analysis of data was done manually using a word processor (Microsoft Word for Windows 2013). One of the participants who provided contact details was emailed with the results and there was agreement on the categories and subcategories: "I strongly agree with these findings".

## Results

Table 2 below displays the characteristics of the individuals who took part in the study. The participants (n = 20) were between 26 and 74 years old (the average age was 42 years); there

**Table 2. Characteristics of participants (n = 20).**

| Characteristics | Category | N (%) |
|---|---|---|
| Gender | Male | 5 (25) |
| | Female | 15 (75) |
| Age | <35 | 6 (30) |
| | 35–50 | 10 (50) |
| | >50 | 4 (20) |
| Education | Secondary | 5 (25) |
| | Technical | 5 (25) |
| | University | 10 (50) |

**Table 3. Categories and subcategories (interview number).**

| Category | Sub-Categories |
|---|---|
| Satisfaction with telemedicine | Improvement of health as a result of telemedicine visits *(1, 6, 9)* |
| | Opportunity to express patients' concerns *(3, 5, 7, 18, 20)* |
| | Telemedicine visit as a follow-up *(8)* |
| | Ensuring continuity of care *(15, 17)* |
| | The physician-patient relationship *(10, 14)* |
| | Convenience *(2, 4, 13, 19)* |
| Dissatisfaction with telemedicine | Lack of physicians' involvement in conducting the telemedicine visit *(16)* |
| | Leaving the patients unsupported *(12)* |
| | Misdiagnosis *(11)* |

were 15 females and 5 males. Five participants had secondary education, ten higher education, and five vocational education. All the participants were of Polish nationality.

The analysis of the content of the transcripts identified the following categories related to how patients understand satisfaction or dissatisfaction with telemedicine: (1) health improvement, (2) opportunity to express patients' concerns, (3) ensuring continuity of care, (4) the doctor-patient relationship, (5) follow-up consultation, (6) convenience, (7) lack of physician's involvement in conducting telemedicine, (8) leaving patients unsupported, (9) misdiagnosis (Table 3).

## 1.1. Improvement of health as a result of a telemedicine visit

Patients were satisfied when their telemedicine consultation was effective. They interpreted this as a consequence of the elimination of the disease symptoms and improvement of their health.

> "Of course, I'm happy with the telemedicine consultation. When I called, I explained why I was calling. The doctor asked more questions and I answered. I talked about additional ailments and was attentively listened to and offered advice. And I got an electronic prescription. I'm happy my symptoms are gone. Telemedicine consultations are definitely effective. If there is a problem with getting to the clinic, it is easier to through it over the phone at my age." (female, 74 years old, secondary education)

Some patients, in addition to improving their health, emphasized that during the telemedicine visit, they were informed that they should make an appointment in person if the symptoms would not subside.

However, they pointed out that in the case of atypical symptoms or symptoms indicative of COVID-19, they would prefer a personal visit with the possibility of auscultation.

> "Because appropriate treatment was recommended, and after my treatment, the symptoms went away. However, I was informed that if the symptoms did not subside, I should make an appointment in person. If it was some unusual infection or with something with symptoms typical of Covid, then I would prefer someone to auscultate me." (female, 32, secondary education)

> "I am happy with the teleconsultation. The doctor explained everything to me. He listened to me carefully, told me what medications I should use and wrote me a prescription, so overall I am satisfied. I believe that the teleconsultation was helpful in my situation as it was

just a common cold because if I had experienced a different disease, I would have preferred to consult a doctor in person. And in such a situation, the teleconsultation was good enough." (female, 26 years old, university education)

## 1.2. Opportunity to express concerns

The satisfaction of some patients with telemedicine visits resulted from the doctor's attitude toward the patient's problem. First of all, patients had the opportunity to talk about their concerns and ask additional questions, and the doctor answered questions and provided information and explanation.

"I was lucky to find myself in the hands of a very dedicated doctor. I told the doctor about my concerns that it could be the same disease I had had earlier because I had been through COVID-19 before. It was explained to me that it was too early for reinfection. I am very pleased with this advice. (. . .) First of all, the doctor showed interest in my problem. She asked me various questions about my health and answered my questions. She told me what medications she was prescribing and how to dose them. All this contributed to the fact that I am satisfied with this teleconsultation." (female, 46 years old, university education)

Patients appreciated the time and freedom provided by the doctor during telemedicine consultation. According to the patients, the physician's involvement in conducting the telemedicine visit did not qualitatively differ from face-to-face consultations.

"I didn't feel any time pressure during the teleconsultation. I was able to take my time and explain what was going on. Especially if the problem concerned a child, I could describe the symptoms in detail. The doctor asked additional questions, (. . .) so it was as detailed as a visit at the doctor's office." (female, 32 years old, university education)

Some participants highlighted that they felt safe during their remote medical consultation due to the in-depth medical history taking that consisted of questions regarding alarm symptoms and elements of physical examination.

"I felt listened to during my teleconsultation. I felt safe. I was asked questions about some elements of the physical examination as well as concerning symptoms such as fainting." (male, 36 years old, university education)

## 1.3. Ensuring continuity of care

The key element influencing the satisfaction of patients with telemedicine visits was constant contact with the family doctor.

"During my teleconsultation, the doctor referred me for a COVID test, which turned out positive. The doctor asked me about saturation, blood pressure, and pulse. My doctor even called me and asked how I was feeling. In my case, a teleconsultation was sufficient. You can easily explain everything during a teleconsultation. For me, it's enough. Speaking with a doctor is the important part." (female, 49 years old, secondary education)

Patients emphasized that satisfaction with telemedicine was a result of a holistic view of the patient. They pointed out that the doctor often called to check the patient's condition.

According to the participants of the study, medical assistance was provided without undue delay, which was particularly important from their point of view.

> "I always feel acknowledged, because we've had teleconsultations with this doctor more than once. The doctor always patiently listens to us and she tries to solve the problem from different angles because the solution isn't obvious. (. . .) The doctor always suggests calling if the symptoms persist and she also calls to ask if everything is okay. I am very pleased with this service. My daughter receives help almost immediately and that's the most important thing." (female, 37 years old, technical education)

### 1.4. The doctor-patient relationship

From the patients' perspective, an important aspect affecting satisfaction with telemedicine was the previously established relationship with the primary physician. Patients trusted doctors who knew them and their medical history.

> "The doctor asked about everything, we stay in touch with her at all times. I have great confidence in her. She knows my children by name, and remembers the course of their illness" (female, 29 years old, university education)

Patients emphasized that they preferred to have their telephone consultation with a well-known family doctor rather than a face-to-face consultation with an unfamiliar physician.

> "People travel a lot nowadays. I can stay in touch with my family doctor when I'm on the other side of the country. It's great. Much better than going to a random physician." (female, 32 years old, university education)

### 1.5. Follow-up consultation

Patients' satisfaction with telemedicine resulted from the possibility of conducting a follow-up medical consultation after a face-to-face visit. From their point of view, avoiding exposure to other infectious diseases in the primary care facility was another advantage of telemedicine.

> "I am satisfied. (. . .) The prescribed medication helped. The doctor listened to me and she had also seen my son before. I don't think I would be happy if this was my first contact with this doctor [via teleconsultation] but we've had contact before so I agreed to a teleconsultation. Telemedicine allows for easier access to a doctor, prescription may be issued. In our case, I was able to have a consultation again over the phone after a personal visit, and didn't have to drag my child to the clinic and expose them to more diseases." (female, 29 years old, university education)

### 1.6 Convenience

From the patients' perspective, telemedicine visits were the preferred method of medical consultation in the case of troublesome ailments that made reaching the family clinic difficult. At the same time, they emphasized that remote consultations could be carried out if the patient's condition did not indicate the need for a physical examination.

"In my view, my teleconsultation was simply more convenient than a personal visit because I was not feeling well. And so I called and I asked about everything. The doctor told me what to do. It seems to me that my health condition didn't require an examination. My cough was not severe. And this teleconsultation was definitely enough for me. Based on my symptoms, the doctor didn't see the need for additional tests, and I agree." (female, 43 years old, technical education)

"Personally, I am satisfied. The doctor was thorough. I was listened to, I could talk about my problem. This is quite a convenient solution for the treatment of certain diseases that do not require an in-person visit." (female, 46 years old, secondary education)

From the patient's perspective, obtaining comprehensive medical advice during a telemedicine visit was possible. Time savings and avoiding inconveniences such as waiting for a personal visit made this option attractive. The final decision on the need to examine the patient in person should be made by the physician.

"Yes, I am satisfied. The problem has been solved. The treatment was effective. I received help. Receiving the prescription and sick leave, all of that could be done via teleconsultation. I think that if the doctor says that an in-person visit is necessary, then we should agree to it, but if not, then there is no point in going there, taking up a queue, and wasting time, as it can be done via teleconsultation." (male, 33, university education)

An important aspect of ensuring patient satisfaction with telemedicine was easier access to medical care compared to a face-to-face visit.

"On the one hand, it was a faster form of contact and with my ailments, it was difficult for me to go to the clinic. I am satisfied with the teleconsultation. When a patient feels bad, it is difficult for them to go to the doctor, or medical advice needs to be given quickly, teleconsultations are appropriate." (male, 30 years old, technical education)

From the patients' perspective, satisfaction with telemedicine visits resulted from the greater availability of medical services and the implementation of the diagnostic and therapeutic process, which allowed for time savings.

"I am satisfied, I got all of the explanations I wanted. Together with the doctor, we decided what tests I should do and what further steps I should take. This is good because it seems to me that it allows for faster access. The doctor could order some tests for me and then, during the in-person visit, I could come back with specific results. Simply put, the treatment faster." (male, 38, technical education)

For some patients, telemedicine visit was a convenient way of obtaining sick leave. However, self-assessment of the patient's health was important, when they themselves believed it not to be a serious disease.

"The teleconsultation was satisfactory. Everything was done as it was supposed to. I could describe my ailments in detail. But to be honest, I wanted to get sick leave from work because I wasn't feeling well, but it didn't look like a serious illness." (male, 42 years old, university education)

## 2. Dissatisfaction with telemedicine

**2.1. Lack of involvement of the doctor in conducting the telemedicine visit.** Some patients highlighted that the telemedicine visit they had received was ineffective. In their understanding, this was due to haste and lack of involvement of the doctor in conducting the medical consultation. In addition, patients emphasized that the physician was not familiar with the patient's medical history and was unable to show empathy during remote consultation, which was particularly important from the patient's point of view.

"I had the impression that my teleconsultation was rushed because people need to talk about their ailments. I don't use it [teleconsultation] unless it's necessary, but the doctor didn't ask me about anything else, he prescribed antibiotics, thank you, goodbye. It seems to me that if the doctor had looked at the previous records kept by the attending physician, he would more or less know just what antibiotic works for me. It is important to have a detailed, empathetic conversation with the patient. Sometimes, the patient may exaggerate in some matters, but it is certain that they don't feel well. Yes, this is probably the most important thing, a certain kind of kindness, and showing such kindness is important when talking to a patient." (female, 64, university education)

**2.2. Leaving the patient unsupported.** Some patients emphasized that during the telemedicine visit, they did not receive sufficient support from physicians. In their understanding, they were left alone with a health problem. In their opinion, due to the COVID-19 pandemic, other medical problems have been pushed to the side.

"At that time, there was only one diagnosis–COVID-19. The doctor said if it got worse, I should call an ambulance. Honestly? It was a nightmare back then. We didn't know what we were dealing with. The doctor said that I should take anti-inflammatory and antipyretic drugs. I called again after some time and said that it was a bit better. Since it was better, that was the end of that; only symptomatic drugs. We were on our own. (or I was on my own). So I don't look back at it with fond memories." (female, 56 years old, secondary education).

**2.3. Misdiagnosis during telemedicine consultations.** Patients' dissatisfaction resulted from the limitations posed by telemedicine visits. Patients were not always able to adequately identify their medical ailments, which may lead to misdiagnosis.

"At first, I was prescribed antibiotics and they did not help. I got in touch with the doctor again and she said it was Omicron. [. . .] That was the limitation of doing it over the phone, when a person feels really bad and is unable to show everything, it's not enough. I finally went and showed myself through the door and "oh, it's angina". The doctor was unable to tell over the phone, so in such situations, a teleconsultation doesn't do the job." (female, 58, university education)

## Discussion

This qualitative research reported patients' experience with telemedicine in primary care settings in the Podlaskie Voivodeship during the COVID-19 pandemic. The main objective was to comprehend the patients' understanding of satisfaction with telemedicine. In our study, the majority of participants responded positively to switching to remote consultations in primary care, which was consistent with the research conducted by The Ministry of Health [14].

The qualitative study, conducted before the COVID-19 pandemic, established the following ways of understanding patients' satisfaction with the care provided by family doctors during face-to-face visits, i.e. good doctor-patient interaction, health improvement, fulfillment of prior expectations, and availability of care [27]. Although the results of the qualitative research were difficult to compare, some topics were common and overlapped. In our research, satisfaction with telemedicine in primary care was defined as health improvement resulting from telemedicine consultation, ensuring continuity of medical care, and the opportunity to express one's concerns. The participants also paid attention to the established doctor-patient relationship.

Pre-pandemic studies on telemedicine utilization among primary care facilities were limited. The research revealed that patients' satisfaction with telemedicine was related to convenience and improved access to medical care, which was also indicated in our study [31–35]. The increased access to medical services compared to face-to-face visits enables a more prompt response to patients' needs and allows for consulting a greater number of patients [36, 37]. The convenience of telemedicine consultation, in terms of saving time, and reducing stress, travel, or employment disruption, may indirectly improve access for individuals with previously low engagement in healthcare [34]. Furthermore, telemedicine creates more equitable access to medical care for those finding face-to-face visits difficult, e.g. people with disabilities or difficulties with arriving at the clinic. It should be emphasized that due to the fact that young patients may overuse telemedicine and certain age groups may need more assistance in participating in a telemedicine consultation, elderly patients may not be able to benefit from telehealth. In the mentioned scenario, telemedicine may even reduce access to medical care.

From the participants' perspective, satisfaction with telemedicine was associated with receiving enough space to express their concerns. Our study confirmed the reports by other authors, that the most significant aspect was receiving telemedicine consultation without the pressure of time, in which patients felt reassured and heard [36]. Some patients indicated that the quality of care during telemedicine consultation did not differ from a face-to-face visit [35, 38]. In previously conducted studies, patients described telemedicine visits to be more personal and focused, and they felt more comfortable communicating during a remote encounter [37]. As a result of telemedicine, the relationship between doctor and patient grows into a more intimate, closer connection regardless of physical distance [39]. In line with the body of research, patients' satisfaction with telemedicine was built on established relationships with their primary care physicians [35–39]. The participants reported that they trusted their primary care physicians and felt comfortable during telemedicine consultations [35]. From the participants' perspective, connecting with a known, trusted doctor was more significant than having a face-to-face visit with an unfamiliar physician. A few patients contradicted these findings and emphasized the insufficient involvement of physicians and lack of empathy during remote consultations, which was particularly important during the COVID-19 pandemic. In the participants' opinion, they were left alone in the face of a new phenomenon that caused great fear. They described concerns about being overlooked during telemedicine visits as they felt the approach was intended to prevent patients from having face-to-face consultations. Therefore, in order to prevent patients from feeling overlooked, physicians should demonstrate an understanding of the patient's medical ailments and clearly explain why a face-to-face consultation is not suitable in those particular cases. Those impersonal consultations were also related to the lack of an established patient-doctor relationship [37]. On the other hand, in the previous study, even when the pre-existing patient-doctor relationship had not been established, telemedicine visits could still prove to be successful if the physicians built a rapport with their patients [36]. Providing 'evidence-based', compassionate care with shared decision making contributes to a strong physician–patient relationship which should be the main goal

of primary care. It is clear that specific communication skills are required for doctors in the environment of telemedicine consultations to provide patient-centered care [40].

In our study, the participants described satisfaction as the effectiveness of their treatment. The sense of health safety among many participants was largely ensured by taking an in-depth medical history. In line with the body of research, it was emphasized that rather than implementing telemedicine consultations for all consultations, a particular focus should be placed on minor conditions where, according to the patients, a physical examination was not required [37]. The participants emphasized that in the event of unknown or unstable conditions, they would prefer to be seen in person and receive a physical examination. In particular cases, the patients found the lack of physical interaction would negatively affect the care received during a telemedicine visit [34]. However, the effects of the lack of face-to-face interaction could be mitigated by using video consultations rather than relying on audio as the only form of telemedicine [34]. Additionally, it was found that participants believed that telemedicine enables physicians to decide if patients require a face-to-face visit, which may reduce unnecessary in-person consultations in primary care [34].

## Strengths and limitations

This is a qualitative study exploring patients' perceptions of telemedicine during the COVID-19 pandemic in greater depth. Our study's greatest strength is that the participants were recruited from different primary care settings and represented a variety of experiences of telemedicine. Although we interviewed a limited number of patients, we reached theoretical saturation in our analyses. There are also limitations to our research. The majority of respondents in our research were women, who are more likely to participate in studies than men. The study was conducted during a global pandemic, when patients may have been reluctant to attend face-to-face visits. Therefore, it cannot be presumed that the reported data represents patient attitudes to telemedicine beyond the pandemic.

## Recommendations for healthcare providers and policy makers

The results of qualitative research reported patients' experience with telemedicine in primary care settings allowed us to develop recommendations for healthcare providers and policymakers.

**Recommendations for healthcare providers.**

1. Empathy and Communication: Train physicians to show empathy and communicate effectively during telemedicine consultations.

2. Clear Information: Provide thorough and clear explanations of treatments, outcomes, and follow-up steps.

3. Flexibility: Offer in-person visits when telemedicine falls short or if patients prefer it.

4. Feedback: Regularly gather and use patient feedback to improve telemedicine services.

5. Physician training in the use of technology: Ensure healthcare professionals are well-trained in telemedicine technologies.

**Recommendations for policy makers.**

1. Patient-Centered Focus: Prioritize patient preferences and choices in telemedicine policies.

2. Innovation: Encourage collaboration between healthcare providers, technology companies, and researchers to continuously innovate and enhance telemedicine platforms. Support initiatives that explore ways to make telemedicine more user-friendly, effective, and accessible.

3. Research Funding: Allocate resources for qualitative research into patient experiences with telemedicine beyond the pandemic, to guide improvements.

## Conclusion

The results of the study revealed patients' in-depth perception of telemedicine. They are also a substantive summary of the implementation period of telemedicine in primary health care during the COVID-19 pandemic. In our research telemedicine met with a positive reception and was recognized by the majority of patients who had experience with it as a valuable channel of contact with a primary care physician. In order to increase the level of patient satisfaction, the focus should be on improving aspects such as physician engagement and showing empathy during telemedicine, as well as providing a full, exhaustive explanation of the treatment process. It is important to provide face-to-face visits when telemedicine consultation does not produce the expected result or if that is the patient's preference. Respecting patient needs and preferences while performing telemedicine visits is the goal of patient-centered care. Further research into patient perceptions and experience with telemedicine in primary care outside the COVID-19 pandemic using qualitative research is required to reveal how patients' perceptions change over time. This will lead to improvement in patient satisfaction and patient-centered care and will enable the implementation of further modern telemedicine solutions in primary care.

## Supporting information

**S1 File. COREQ guidelines.** Consolidated criteria for reporting qualitative studies (COREQ): 32-item checklist.
(DOCX)

## Author Contributions

**Conceptualization:** Karolina Pogorzelska, Slawomir Chlabicz.

**Data curation:** Karolina Pogorzelska, Ludmila Marcinowicz.

**Formal analysis:** Karolina Pogorzelska, Ludmila Marcinowicz.

**Funding acquisition:** Karolina Pogorzelska, Slawomir Chlabicz.

**Investigation:** Karolina Pogorzelska.

**Methodology:** Karolina Pogorzelska, Ludmila Marcinowicz, Slawomir Chlabicz.

**Project administration:** Karolina Pogorzelska, Slawomir Chlabicz.

**Resources:** Karolina Pogorzelska.

**Supervision:** Ludmila Marcinowicz, Slawomir Chlabicz.

**Validation:** Slawomir Chlabicz.

**Writing – original draft:** Karolina Pogorzelska.

**Writing – review & editing:** Ludmila Marcinowicz, Slawomir Chlabicz.

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
