## [Decision Letter · Decision Letter 0]

24 Jul 2023

PONE-D-23-18556Understanding satisfaction and dissatisfaction of patients with telemedicine during the COVID-19 pandemic: an exploratory qualitative study in primary care.PLOS ONE

Dear Dr. Pogorzelska,

Thank you for submitting your manuscript to PLOS ONE. After careful consideration, we feel that it has merit but does not fully meet PLOS ONE’s publication criteria as it currently stands. Therefore, we invite you to submit a revised version of the manuscript that addresses the points raised during the review process.

We look forward to receiving your revised manuscript.

Kind regards,

Mohammad Nusair, Ph.D

Academic Editor

PLOS ONE

Journal Requirements:

Additional Editor Comments:

I highly encourage the authors to revise the manuscript according to the reviewers's comments and give careful attention to the methods section. I have additional comments for the authors to consider:a) please revise the methods section in the abstract to give the readers a clear summary of your methodsb) how were the 24 patients selected? was it a random selection?c) participants were invited to take part in the study over the phone, but how did you conduct the interviews? was it a telephone interview?d) some of your interview questions were closed ended. this is very unusual for studies of qualitative nature.e) reviewer 2 commented "I was not able to sufficiently judge if the statistical analysis was appropriate given the missing information as explained above."i believe the reviewer meant "content analysis" instead of "statistical" and I agree with the reviewer comment. Please give the reader more information on how content analysis was performed.f) your quotes start with  „ and ends with " . I am not sure if that is a technical error but please ensure that quotes start with " and end with "

Reviewers' comments:

Reviewer's Responses to Questions

**Comments to the Author**

1. Is the manuscript technically sound, and do the data support the conclusions?

Reviewer #1: Yes

Reviewer #2: Partly

2. Has the statistical analysis been performed appropriately and rigorously? 

Reviewer #1: N/A

Reviewer #2: I Don't Know

3. Have the authors made all data underlying the findings in their manuscript fully available?

Reviewer #1: Yes

Reviewer #2: No

4. Is the manuscript presented in an intelligible fashion and written in standard English?

Reviewer #1: Yes

Reviewer #2: Yes

5. Review Comments to the Author

Reviewer #1: Dear Authors,

Your article is novel and worth publication, however, I have few comments (as below) that I would like you to consider before your article is ready to be published:

A. Introduction:

1. Last paragraph before method section: please add a reference to this statement as you said in the literature? without referencing the literature "In the literature, patients’ perceptions of telemedicine are often used as a measurement of patient satisfaction. ". you can select from the below references to support your statement:

* Khasawneh RA, Al-Shatnawi SF, Alhamad H, Rahhal D. General Public Perceptions and Perceived Barriers Toward the Use of Telehealth: A Cross-Sectional Study from Jordan. Telemedicine and e-Health. 2023 Feb 17.

* Mason AN, Brown M, Mason K. Telemedicine patient satisfaction dimensions moderated by patient demographics. InHealthcare 2022 Jun 1 (Vol. 10, No. 6, p. 1029). MDPI.

* Nguyen M, Waller M, Pandya A, Portnoy J. A review of patient and provider satisfaction with telemedicine. Current allergy and asthma reports. 2020 Nov;20:1-7.

* Noceda AV, Acierto LM, Bertiz MC, Dionisio DE, Laurito CB, Sanchez GA, Loreche AM. Patient satisfaction with telemedicine in the Philippines during the COVID-19 pandemic: a mixed methods study. BMC Health Services Research. 2023 Dec;23(1):1-2..

B. Methods:

1. Design: the first paragraph: "The interviews were performed until unique categories were no longer

identified and data saturation was achieved". Please explain here more on how you achieve data saturation.

2. Table 1: the last three questions in this table are pure closed ended questions? which can be used in qualitative study only if a follow up question is used such as "Please explain more". Kindly, add "Please explain more" after each of these three questions.

C. Discussion:

1. Last paragraph: "The majority of respondents in our research were women, who are more likely to participate in studies than men. The study was conducted during a global pandemic, when patients may have been reluctant to attend face-to-face visits.". Please add "and these would be a limitations to our study".

2. After the last paragraph (Just before the conclusion section): Please add one paragraph where you clearly translate the major findings of your study as a recommendation to policymakers and healthcare providers.

D. Conclusion: Very well written. Thank you.

Kind regards.

Reviewer #2: July 2023

Dear Editor-in-chief,

I would like to cordially thank you for giving me the chance to review the interesting manuscript entitled “Understanding satisfaction and dissatisfaction of patients with telemedicine during the COVID-19 pandemic: an exploratory qualitative study in primary care” which reported the results of a qualitative study conducted in Poland. The authors did a good job describing the research in general, however, I do have the following recommendation to improve the scientific soundness of the reported research:

Abstract

I do recommend elaborating more on the rationale of the study since the authors do have a window to do so. Please include more in the background section to highlight the importance of adoption of telemedicine during COVID-19. The authors are also advised to add details regarding the methodology of the study; it was so concise and missed major information.

Introduction

The authors discussed COVID-19 in depth and add enough details about telemedicine although it is the core of the study.

- I do suggest focusing more on telemedicine; add definition, modalities of implementation, how well is Poland ready for adoption in regard to infrastructure and familiarity of both healthcare providers and patients.

- Besides, I do suggest adding more about the rationale of the study; why is it important to study telemedicine and patient satisfaction dissatisfaction although the pandemic is now over? The authors need to convince the audience about their point of view.

Methods/Results

The methods section missed major pieces of information rendering the study heard to replicate.

- I highly encourage the authors to adopt the Consolidated Criteria for Reporting Qualitative studies (COREQ) checklist or Standards for Reporting Qualitative Research (SRQR) Reporting checklist for qualitative study to add all missing information as it appears on the checklist in order to enhance the transparency of reporting of the qualitative study.

- I was not able to sufficiently judge if the statistical analysis was appropriate given the missing information as explained above.

- The authors did not clarify whether the raw data related to the research will be publicly available or restricted.

At the end, I would like to thank you again for giving me the chance to review this manuscript.

Sincerely,

6. PLOS authors have the option to publish the peer review history of their article (what does this mean?). If published, this will include your full peer review and any attached files.

Reviewer #1: **Yes: **Dr. Hamza Alhamad

Reviewer #2: No

---

## [Author Response · Author response to Decision Letter 0]

20 Sep 2023

Response to Reviewer 1

Dear Reviewer,

We greatly appreciate the time you dedicated to reading our paper and providing us with your suggestions. The feedback you provided has proven invaluable in guiding us toward enhancing the quality of our manuscript.

Our next step is to attend to your comments.

A. Introduction

We extend our gratitude for the reference suggestions, which have been duly credited within our article. A portion of the text referencing the cited article is located on page 7 of our manuscript.

B. Methods

1. Design.

In the "Methods: Design" section, we elucidated the process through which data saturation was achieved. A section of the text discussing the achievement of data saturation can be found on page 7.

2. Table 1.

We have duly corrected the last three questions within the table as per your recommendations. Table 1 is situated on page 8 of our manuscript.

C. Discussion 

1. Last paragraph 

As advised, we have included a sentence in the final paragraph addressing the limitations of our study. 

2. After the last paragraph 

Our manuscript now also features recommendations tailored for policymakers and healthcare providers, augmenting its practical relevance. 

The recommendations can be found on page 23 of our manuscript.

Notable modifications have been highlighted in red throughout the text.

We truly hope the paper will be found suitable for publication.

Best regards,

Karolina Pogorzelska

Response to Reviewer 2

Dear Reviewer,

We greatly appreciate the time you dedicated to reading our paper and providing us with your suggestions. The feedback you provided has proven invaluable in guiding us toward enhancing the quality of our manuscript.

Our next step is to attend to your comments.

A. Abstract

As per your suggestion, the methods section within the abstract succinctly summarizes the methodologies used in our qualitative research. Furthermore, we have added missing information to the background section.

B. Introduction

In the introduction section, we have delved extensively into the topic of telemedicine and its adoption within Poland. An extended introduction, containing additional information, is available on pages 3-7 of our manuscript.

C. Methods/Results

Acting upon your valuable suggestion, we have integrated the Consolidated Criteria for Reporting Qualitative Studies (COREQ) checklist. 

This addition aims to incorporate any previously omitted information, aligning with the checklist's criteria to enhance the transparency of our qualitative study's reporting. 

Substantial missing details have been seamlessly incorporated into the methods section.

All relevant data are within the manuscript and its Supporting Information files. Data cannot be shared publicity as participants did not give consent for their transcripts to be shared in the public domain. Data are available for selected researchers from the Medical University of Bialystok, who meet the criteria for access to confidential data. Requests for access to the underlying data may be directed to the Bioethics Committee of the Medical University of Bialystok komisjabioetyczna@umb.edu.pl

Notable modifications have been highlighted in red throughout the text.

We truly hope the paper will be found suitable for publication.

Best regards,

Karolina Pogorzelska

---

## [Decision Letter · Decision Letter 1]

5 Oct 2023

Understanding satisfaction and dissatisfaction of patients with telemedicine during the COVID-19 pandemic: an exploratory qualitative study in primary care.

PONE-D-23-18556R1

Dear Dr. Pogorzelska,

We’re pleased to inform you that your manuscript has been judged scientifically suitable for publication and will be formally accepted for publication once it meets all outstanding technical requirements.

Kind regards,

Mohammad Nusair, Ph.D

Academic Editor

PLOS ONE

Additional Editor Comments (optional):

Reviewers' comments:

Reviewer's Responses to Questions

**Comments to the Author**

1. If the authors have adequately addressed your comments raised in a previous round of review and you feel that this manuscript is now acceptable for publication, you may indicate that here to bypass the “Comments to the Author” section, enter your conflict of interest statement in the “Confidential to Editor” section, and submit your "Accept" recommendation.

Reviewer #1: All comments have been addressed

Reviewer #2: All comments have been addressed

2. Is the manuscript technically sound, and do the data support the conclusions?

Reviewer #1: Yes

Reviewer #2: Yes

3. Has the statistical analysis been performed appropriately and rigorously? 

Reviewer #1: Yes

Reviewer #2: Yes

4. Have the authors made all data underlying the findings in their manuscript fully available?

Reviewer #1: Yes

Reviewer #2: Yes

5. Is the manuscript presented in an intelligible fashion and written in standard English?

Reviewer #1: Yes

Reviewer #2: Yes

6. Review Comments to the Author

Reviewer #1: Dear Author,

Thank you for submitting the revised version and addressing my comments. I think the article now is in a good shape to be published

Decision is "Accept".

Reviewer #2: I would like to commend the authors for their exceptional responsiveness and receptiveness to my comments. They acknowledged my feedback, resulting in a manuscript that has significantly benefited from their diligence and commitment to excellence. I am thoroughly impressed by the improvement in the manuscript and how they addressed all points as raised by the reviewers and the respected editor. Great Work!

7. PLOS authors have the option to publish the peer review history of their article (what does this mean?). If published, this will include your full peer review and any attached files.

Reviewer #1: No

Reviewer #2: No

---

## [Editor Report · Acceptance letter]

9 Oct 2023

PONE-D-23-18556R1 

Understanding satisfaction and dissatisfaction of patients with telemedicine during the COVID-19 pandemic: an exploratory qualitative study in primary care. 

Dear Dr. Pogorzelska:

I'm pleased to inform you that your manuscript has been deemed suitable for publication in PLOS ONE. Congratulations! Your manuscript is now with our production department. 

Kind regards, 

on behalf of

Dr. Mohammad Nusair 

Academic Editor

PLOS ONE